# Evaluation of Reference Genes for Normalizing RT-qPCR and Analysis of the Expression Patterns of *WRKY1* Transcription Factor and Rhynchophylline Biosynthesis-Related Genes in *Uncaria rhynchophylla*

**DOI:** 10.3390/ijms242216330

**Published:** 2023-11-15

**Authors:** Detian Mu, Yingying Shao, Jialong He, Lina Zhu, Deyou Qiu, Iain W. Wilson, Yao Zhang, Limei Pan, Yu Zhou, Ying Lu, Qi Tang

**Affiliations:** 1College of Horticulture, Hunan Agricultural University, Changsha 410128, China; mudetian12580@163.com (D.M.); 16680541946@163.com (Y.S.); 18008453667@139.com (J.H.); ningzby@163.com (Y.Z.); 18574824089@163.com (Y.Z.); 2State Key Laboratory of Tree Genetics and Breeding, Research Institute of Forestry, Chinese Academy of Forestry, Beijing 100091, China; qiudy@caf.ac.cn; 3Commonwealth Scientific and Industrial Research Organisation (CSIRO) Agriculture and Food, Canberra, ACT 2601, Australia; 4Key Laboratory of Guangxi for High-Quality Formation and Utilization of Dai-di Herbs, Guangxi Botanical Garden of Medicinal Plants, Nanning 530023, China; plimei2006@163.com; 5Hunan Provincial Key Laboratory for Synthetic Biology of Traditional Chinese Medicine, Hunan University of Medicine, Changsha 410208, China

**Keywords:** *Uncaria rhynchophylla*, RT-qPCR, reference gene, terpenoid indole alkaloids, expression pattern

## Abstract

*Uncaria rhynchophylla* (Miq.) Miq. ex Havil, a traditional medicinal herb, is enriched with several pharmacologically active terpenoid indole alkaloids (TIAs). At present, no method has been reported that can comprehensively select and evaluate the appropriate reference genes for gene expression analysis, especially the transcription factors and key enzyme genes involved in the biosynthesis pathway of TIAs in *U. rhynchophylla*. Reverse transcription quantitative PCR (RT-qPCR) is currently the most common method for detecting gene expression levels due to its high sensitivity, specificity, reproducibility, and ease of use. However, this methodology is dependent on selecting an optimal reference gene to accurately normalize the RT-qPCR results. Ten candidate reference genes, which are homologues of genes used in other plant species and are common reference genes, were used to evaluate the expression stability under three stress-related experimental treatments (methyl jasmonate, ethylene, and low temperature) using multiple stability analysis methodologies. The results showed that, among the candidate reference genes, S-adenosylmethionine decarboxylase (*SAM*) exhibited a higher expression stability under the experimental conditions tested. Using *SAM* as a reference gene, the expression profiles of 14 genes for key TIA enzymes and a *WRKY1* transcription factor were examined under three experimental stress treatments that affect the accumulation of TIAs in *U. rhynchophylla*. The expression pattern of *WRKY1* was similar to that of tryptophan decarboxylase (*TDC*) under ETH treatment. This research is the first to report the stability of reference genes in *U. rhynchophylla* and provides an important foundation for future gene expression analyses in *U. rhynchophylla.* The RT-qPCR results indicate that the expression of *WRKY1* is similar to that of *TDC* under ETH treatment. It may coordinate the expression of *TDC*, providing a possible method to enhance alkaloid production in the future through synthetic biology.

## 1. Introduction

*U*. *rhynchophylla* (Miq.) Miq. ex Havil is an evergreen liana belonging to the *Rubiaceae* family [1] that has been used to treat convulsions, hypertension, epilepsy, eclampsia, and other cerebral diseases for a long time as a traditional Chinese herbal medicine [2]. *Uncaria* contains alkaloids, flavonoids, triterpenoids, and other active ingredients [3,4,5]. The main active ingredients in *U. rhynchophylla* are terpenoid indole alkaloids (TIAs), such as rhynchophylla (RIN) and isorhynchophylla (IRN), which exert significant effects in the treatment of Alzheimer’s disease by acting on multiple pathological processes [6]. Rhynchophylline is known to show antihypertensive activity; it can attenuate AngII-induced elevation of blood pressure and protect against myocardial ischemia-reperfusion injury [7,8]. *U. rhynchophylla* TIA pathways have been the focus of many studies due to their potential value in the pharmaceutical industry.

Although the active ingredients of *U. rhynchophylla* exhibit many therapeutic properties, the major challenge for their utilization by the pharmaceutical industry is the low production rate of these alkaloids [9] and the fact that genetic improvement through traditional breeding is long and costly. TIAs, such as RIN and IRN, are thought to be part of the plant chemical defense system against pests, and stress-signal-related molecules or pathogen-derived elicitors are often used to induce the accumulation of these precious pharmaceutic active metabolites [10]. However, with the advent of synthetic biology, research is now centered on the molecular elucidation of the important biosynthesis pathways of medicinal plants such as *U. rhynchophylla* to identify key genes that synthesize and control the alkaloid content, which would allow these alkaloid pathways to be established and expressed in more genetically amenable systems [11].

TIA biosynthesis involves multiple complex metabolic pathways. All TIAs are derived from tryptamine and secologanin, which are derived from the tryptophan and iridoid terpene biosynthesis pathways, respectively. The biosynthetic pathways for RIN and IRN are shown in Figure 1. In the first committed step of iridoid terpene biosynthesis, geranyl-PP is converted to geraniol by geraniol synthase (GES), and a variety of functional enzymes participate in oxidation, reduction, glycosylation, and methylation reactions. Finally, secologanin is generated by the oxidative cleavage of loganin by the enzyme secologanin synthase (SLS). In the tryptamine pathway, chorismate is catalyzed by a variety of enzymes to generate L-tryptophan, which is catalyzed by tryptophan decarboxylase (TDC). Thus, the condensation of secologanin and tryptamine, catalyzed by strictosidine synthase (STR), forms strictosidine. Finally, the compound is desugarized by strictosidine-*β*-D-glucosidase (SGD). However, detailed studies on the molecular structures and related biosynthesis pathways of TIAs are sparse.

With the development of high-throughput sequencing technology, many biosynthesis pathways of medicinal plants have been elucidated. The elucidation of secondary metabolism pathways often requires the study of key enzyme genes along pathways. Gene expression analysis by RT-qPCR is a commonly used method to study gene function [12]. In *Euodia rutaecarpa*, the expression patterns of *HMGCR*, *SQE*, and *CYP450* calculated by screened reference genes were consistent with the content of limonin, thus implying their potential involvement in the limonin biosynthesis of *E. rutaecarpa*. Furthermore, qRT–PCR was used to analyze the expression patterns of 15 genes involved in the terpenoid indole alkaloids pathway using stable reference genes, and three key enzyme genes were found to have strong correlations with the gelsenicine in *Gelsemium elegans* [13,14]. qRT–PCR has high sensitivity, high specificity, and simple operation [15]. However, the accuracy and reliability of the results are dependent on several factors, including the quality of the RNA, the efficiency of the reverse transcription process, the specificity of the primers, and the stable expression of reference genes that are used to normalize the RNA levels between samples [16,17]. Recent studies have shown that the stable expression of reference genes in different tissues, organs, developmental periods, and under different treatment conditions is rare or nonexistent [18,19]. Traditional reference genes commonly used for RT-PCR in plants include glyceraldehyde-3-phosphate dehydrogenase (*GAPDH*), actin (*ACT*), elongation factor 1α (*EF-1α*), 18S ribosomal RNA (*18S rRNA*), α-tubulin (*TUA*), and S-adenosylmethionine decarboxylase (*SAM*), but they are not always stably expressed in all tissues or conditions [18,19]. Thus, having a systematic experimental design for the selection of stable reference genes is an important first step towards the application of RT-PCR in new plant species. In *Catharanthus roseus*, two important plant hormones, ETH and MeJA, were confirmed to be involved in the regulation of TIA biosynthesis. Many genes involved in this pathway and transcriptional regulators have been reported to be regulated by these plant hormones, which indicates that Catharanthus TIA biosynthesis is involved in the MeJA and ETH signal transduction pathways [20]. Furthermore, in *Rauvolfia verticillata,* the expression of *RvTDC* was found to be slightly upregulated by MeJA, and the expression level of *RvTDC* was positively correlated with the content of ajmalicine [21]. Obtaining a stable reference gene is the first step in studying the expression pattern of key enzyme genes and transcription factors in the pathway under different plant hormone treatments and abiotic stresses. Therefore, to better understand the TIA pathway, the first step is to identify the gene expression patterns using screened stable reference genes under different abiotic and biotic stresses.

In this research, ten candidate reference genes were selected from genome of *U. rhynchophylla* (the genome size was 627 Mb, the contig N50 was 1.8 Mb, and 46,909 genes were identified) to explore which reference genes could be stably expressed under exposure to three treatments. According to the most stable reference gene, the expression patterns of the TIA pathway genes and the *UrWRKY1* transcription factor, which shows the highest homology with *CrWRKY1* and has been confirmed to positively regulate TIAs in *C. roseus*, were detected under the three treatments [22]. Our work lays the foundation for future functional analyses of the key genes associated with the biosynthesis and regulation of the TIA pathways.

## 2. Results

### 2.1. Screening for Candidate Reference Genes and Primer-Specific Analysis

In this study, ten candidate reference genes were selected based on normalization in RT-qPCR analyses in other plant species and expression profiles based on our genome Nr database of *U. rhynchophylla* (unpublished). The amplification efficiency (E) and correlation coefficients (R^2^) were calculated by a standard curve with a series of five different cDNA dilutions. The candidate reference gene abbreviations and names, primer sequences, amplification efficiency (E), and correlation coefficient (R^2^) are listed in Table 1. The primer specificity of the candidate reference genes was checked by RT-qPCR. As shown in Appendix A, the melting curves of the candidate reference genes all showed a single and smooth melting peak, and the repeatability of the PCR amplification curves was good. The amplification efficiency ranged from 99.3% for *ACT6* to 105% for *EF1-β*, and the correlation coefficient of the 10 candidate reference genes was above 0.99.

### 2.2. Expression Abundance of the Candidate Reference Genes

The Ct value directly reflects the gene expression abundance (when amplicons exhibit similar reaction efficiencies), and the smaller the Ct value, the higher the gene expression abundance. To evaluate the stability of the reference genes in all treated experimental samples, the expression abundances of the ten candidate reference genes were measured by the mean Ct values. As indicated by Figure 2, the Ct values ranged from 8.39 for *18S* to 27.55 for *GAPDH* under the three treatments in all samples, and most reference genes had values distributed between 22 and 25. The *18S* gene and *GAPDH* exhibited the lowest and highest Ct values, respectively, indicating that *18S* showed the highest expression in all samples, whereas *GAPDH* showed the lowest expression in all samples. The box plot not only displays the expression abundance of the reference gene but also provides an indication of the variation in its expression, because each reference gene had different coefficients of variation, and lower values represent less variability. *SAM* showed the least variability, with an SD value of 0.17, followed by *CYP*, *PP2A,* and *18S. GAPDH* showed the highest mean Ct value (25.84 ± 0.86). However, to ensure that the reference gene was stable, its expression was analyzed using different algorithms.

### 2.3. Expression Stability Analysis of Candidate Reference Genes

TIAs, such as RIN and IRN production, are known to respond to pest- and stress-signal-related molecules. Therefore, when selecting a gene with stable gene expression, it is also necessary to perform the analysis under relevant abiotic stress conditions. To gauge a gene’s potential as a housekeeping gene for analyzing genes involved in TIA pathways, three stress-related experiments were performed involving low temperatures, ETH, and MeJA to ensure that expression was not affected by these conditions. To analyze the reference genes under these three treatments, four algorithms, GeNorm (https://genorm.cmgg.be/, accessed on 27 August 2022), NormFinder (https://www.moma.dk/normfinder-software/, accessed on 27 August 2022), Bestkeeper (https://www.gene-quantification.com/bestkeeper.html, accessed on 27 August 2022), and Delta Ct, were used to verify their expression stabilities. The RefFinder online website was used to systematically select the optimal reference genes under all of the different experimental conditions.

#### 2.3.1. GeNorm Analysis

For GeNrom, the original Ct values of the genes were converted into the relative expression values *Q*, and then the M values were determined [23]. The M value reflects the expression stability, in which a low M value indicates a higher expression stability. As shown in Figure 3, the candidate reference genes showed different stabilities under the three treatments. The M values of *SAM* and *PP2A* were the smallest in the dataset (collection of three treatments) and thus were the most stable of the genes under the three treatments. *EF-1β* and *18S* also showed good stability levels under MeJA. The *cdc73* gene was as stable as *SAM* under ETH. The stability of *TUA* was lower than that of the other candidate reference genes under the three treatments, indicating that this gene was not useful for normalization purposes in this tissue under these conditions. GeNorm can also be used to determine the appropriate number of reference genes by calculating the pairwise variation (Vn/n + 1). When Vn/Vn + 1 < 0.15, the optimal number of reference genes is n; otherwise, the number is n + 1. In this study, all results for pairwise variation under the three treatments calculated by GeNorm were less than 0.15 (Figure 4), suggesting that two reference genes can be used for normalization.

#### 2.3.2. NormFinder Analysis

NormFinder ranks the expression stability of the reference genes by calculating the average paired variation of one gene relative to other candidate reference genes. The smaller the stability value is, the more suitable the gene is as a reference [24]. Table 2 shows the rankings of 10 candidate genes as calculated by the NormFinder algorithm. Among the MeJA, ETH, ‘control’, and ‘total’ treatment subsets, *SAM* was the most stable with the highest score. Under the low-temperature treatment (0.104), *PP2A* was identified as the most stable gene, and *SAM* (0.260) was the second-most stable. Similar to the GeNorm analysis, *TUA* showed the least stable expression.

#### 2.3.3. BestKeeper Analysis

BestKeeper (https://www.gene-quantification.com/bestkeeper.html, accessed on 27 August 2022.), unlike GeNorm and NormFinder, can directly analyze the Ct values obtained by RT-qPCR and calculate the standard deviation (SD) and coefficient of variation (CV) of the Ct values to determine the expression stability of the candidate reference genes [25]. The software uses SD = 1 as the threshold value; if SD > 1, the expression is unstable and not suitable for normalization. However, the smaller the SD is, the more stable the expression of the gene is. The CV reflects the variation in the expression levels of the genes in different treatment groups, and the smaller the CV is, the more stable the expression is. The results of the BestKeeper analysis are shown in Table 3. In the ETH, low temperature, ‘control’, and ‘total’ treatment subsets, *SAM* was ranked as the most stable. Under the MeJA treatment, *ACT6* (1.32 ± 0.31) was the most stable, followed by *cdc73* (1.40 ± 0.35), with *SAM* (1.48 ± 0.33) being the third-most stable. The least stable reference genes calculated by BestKeeper were *TUA* and *18S*. Genes with SD > 1, such as *TUA*, had low expression stability under our conditions and therefore were not suitable for normalization.

#### 2.3.4. Delta Ct Analysis

The gene expression stability was assessed by calculating the mean standard deviation (SD) of the Ct values of the ten candidate reference genes, where the smaller the mean SD is, the higher the expression stability is. As shown in Appendix A, the results are consistent with those from the previous analysis. Compared to the other candidates, *SAM* was found to be more stable in the three treatment groups, and *TUA* was the least stable gene under all experimental conditions. Hence, *SAM* was found to be the most qualified gene for normalization under the conditions tested.

#### 2.3.5. RefFinder Analysis

RefFinder analysis is a method that is used to evaluate the stability comprehensively. To achieve this, RefFinder assigns appropriate weights to the results of each candidate reference gene in each program and calculates the geometric mean of the stability value weights of all algorithms to gain the overall ranking. The RefFinder analysis method avoids the one-sidedness of individual algorithms and provides a comprehensive analysis of the expression stability of the candidate reference genes. The smaller the stability value is, the better the stability of the gene is. As shown in Appendix A, *SAM* was the most stable under all treatments except for MeJA (in which it was fourth), consistent with the results of the GeNorm, NormFinder, Bestkeeper, and Delta Ct algorithms. According to all of the above statistical methods, although *SAM* could not rank at the top in every dataset, *SAM* ranked first for the total of all datasets (collection of the three treatments). Therefore, it was selected as the primary reference gene. In addition, according to GeNorm, two reference genes were applied for normalization, and the *cdc73* reference gene ranked behind *SAM* in almost all algorithms. So, *SAM* and *cdc73* as the most suitable reference genes for analyzing the expression patterns of the pathway genes and *WRKY1* under the three treatments.

### 2.4. Expression Patterns of WRKY1 and Selected Genes Involved in the TIA Pathway

By screening many candidate TIA biosynthetic genes by the “genome + transcriptome + metabolome” coexpression association analysis, this model is highly predictable for the screening of multiple genes or supergene family candidates and can greatly reduce the scope of subsequent gene research. We identified 22 types of genes in the biosynthesis pathway of *Siraitia grosvenorii* Mogroside V using transcriptome data and screened seven *CYP450* and five *UDPG* candidate genes from 80 *CYP450* and 90 *UPDG* genes by using a combined “transcriptome + expression profile + content” coexpression analysis [26]. One or two of these genes were validated by the prokaryotic expression and eukaryotic expression in yeast [27,28]. Coexpression pattern screening of the priority candidate genes is a significant reference for improving the efficiency of screening the supergene family target genes of other species. Taking the *STR* gene as an example (Appendix A), a total of 22 *STR* candidate genes were identified from the genome, and by the transcriptome and content coexpression analysis, two sequences were found to be clustered with the RIN and IRN content, from which the *g3541* gene was first selected as the focus sequence for the qRT-PCR analysis because of the nearest clustering relationship and full-length gene. As previously reported, the overexpression of *CrWRKY1* in *C. roseus* hairy roots upregulated several key genes involved in the TIA pathway and increased the accumulation of serpentine compared to the control. The sequence of *CrWRKY1* was blasted with our *UrWRKY* sequences. *UrWRKY1* shows the highest homology with *CrWRKY1*. Therefore, according to speculation, UrWRKY1 may perform a similar function, and it may also regulate the TIA biosynthesis pathway.

From the Genome Nr database, only one gene was found to be annotated as *WRKY1*. *SAM* and *cdc73* were used as the reference genes to identify and validate *WRKY1* expression and potential key TIA biosynthetic genes in which the application of MeJA, ETH, and cold stress altered the gene expression. We performed qRT-PCR on leaves from tissue culture seedlings that were treated by MeJA and ETH at 0, 1, 2, 4, 8, and 24 h and low-temperature treatments at 0, 4, 8, 12, 24, and 48 h. Under the MeJA treatment, *WRKY1* was significantly upregulated at 4 h (Figure 5a), started to decrease by 8 h, and returned to near to the original expression levels by 24 h. The expression patterns of *TDC* and *SGD* were similar to that of *WRKY1*, especially the expression of *TDC* in response to MeJA at 4 h, which increased by 3.4-fold compared to that at 0 h. The expression levels of some genes, such as *STR* and *SLS*, which were upregulated at 8 h, remained higher than those of untreated genes at 24 h. *G8H* gene expression started to decrease at 1 h after MeJA treatment, was the lowest after 4 h of treatment, and then gradually returned to original levels. As shown in Figure 5b, *WRKY1* also displayed significant changes in expression in response to ETH, which was similar to its response to MeJA. Compared to that at 0 h, *WRKY1* was upregulated by 3-fold by 4 h and then started to decrease by 8 h. The expression of *TDC* was similar to the expression pattern of *WRKY1*; the expression levels of these two genes significantly increased 1 h after treatment, exhibited the highest gene expression after 4 h of treatment, and then slowly declined. The *SGD* gene significantly increased at 2 h, and the expression peaked at 4 h, decreased significantly at 8 h, and was the lowest 24 h after treatment. Furthermore, the expression of *STR* significantly increased at 4 h, peaked at 8 h, and then returned to the lowest initial expression level at 24 h after treatment. The cold stress treatment differed from the other two treatments (Figure 5c). *WRKY1* transcript levels remained lower until 24 h and then rose at 48 h. Among these genes, only *TDC* exhibited an expression pattern similar to that of *WRKY1*. For most genes, their expression levels increased by 4 h and then decreased.

Overall, the expression levels of *WRKY1* and *TDC* increased significantly at 2 h, peaked at 4 h compared with CK, and then showed a downwards trend 24 h after ETH treatment. Under ETH treatment, the changes in the gene expression levels of *TDC* and *WRKY1* basically showed the same trend. This could imply that *WRKY1* is possibly involved in coordinating its expression, which could regulate the TIA biosynthesis pathway.

## 3. Discussion

In recent decades, the protective effects of *U. rhynchophylla* and its major components, TIAs, on the central nervous system and the treatment of cardiovascular diseases have become a major research focus [29,30]. As the major active therapeutic ingredients derived from *U. rhynchophylla* are RIN and IRN, understanding and identifying the unique genes involved in their biosynthesis and regulation are of great significance [31]. Therefore, the ability to quantitate the expression of the known and candidate genes associated with TIA biosynthesis in *U. rhynchophylla* with the specificity, sensitivity, and ease that RT-qPCR provides has become important. However, this methodology is dependent on the knowledge of stable reference genes for normalization, which requires careful empirical determination in the tissues and conditions relevant for the genes and species investigated.

Although the medicinally active content in the leaves of tissue culture seedlings is lower than that in naturally occurring *U. rhynchophylla* plants, it is a more renewable and sustainable resource and can produce a higher biomass, making it an attractive source tissue for RIN and IRN [32,33]. Therefore, our research focused on tissue culture seedling leaves to evaluate the stability of the reference genes for this research. The stability of the reference genes was analyzed under the stress-related signaling molecules MeJA, ETH, and cold stress, which are known to affect TIA production. Cold stress is also a significant threat to plant productivity and impacts the plant distribution and crop production, particularly when it occurs during the growth phase [34].

Several previous studies have indicated that the expression levels of many traditional reference genes can vary among different species and tissues and change under different experimental conditions for the same species [35,36,37,38]. At present, no report has identified a gene with a stable gene expression profile under different stress treatments suitable for RT-pPCR normalization in *U. rhynchophylla*. In this research, we studied ten potential candidate reference genes that were selected based on gene homologues previously used as normalization genes in other plant species. These include protein synthesis (*18S*), the organelle skeleton (*ACT*, *TUA*), biological metabolic processes (*SAM*, *EF-1β*, *PP2A*, *GAPDH*), protein folding (*CYP*), and the regulation of plant growth (*cdc73*, *PAL*). In *Schima superba*, *SsuACT* was the most stable reference gene, and *SsuACT + SsuRIB* were the best parameter combination for the relative expression calculation for different tissues [39]. In *Brachypodium distachyon*, the expression of *SAM* ranked as the most stable in plants grown under various environmental stresses (high salt/drought), and *GAPDH* also plays a housekeeping role in the growth and development of *Brachypodium* [40]. In *orchardgrass*, *ACT* and *CYP* were determined to be the best reference genes for ABA studies [41]. In Tibetan hulless barley, *TUA* and *EF-1β* were the most suitable reference genes under cold stress, and *ACT* was the most stable under drought stress [42]. In *Baphicacanthus cusia*, *18S* was found to be the most stable gene under exposure to ultraviolet irradiation and hormonal stimuli (MeJA and ABA) [43]. In *Sinobambusa tootsik f. luteoloalbostriata*, *SAM* was the most stable internal reference gene among the four leaf colors [44].

The raw Ct values of the candidate reference genes are a direct readout of the expression levels: the lower the Ct value is, the higher the gene expression level is, and a narrow Ct variation range implies high stability [45,46]. As shown in Figure 1, the Ct values of the candidate reference genes ranged from 8.4 to 27.6 under our experimental conditions, and most of them were distributed between 22 and 25. Moreover, wide Ct variation ranges were observed for *EF-1β* and *GAPDH*, indicating that they are not suitable as reference genes for our work. To comprehensively evaluate and compare the expression stability levels of the candidate reference genes, different software and algorithms (GeNorm, NormFinder, Bestkeeper, Delta Ct) were used. According to our overall analysis, *SAM* ranked as the most stable, but there were some differences between analyses. For example, under cold stress conditions, *SAM* was ranked at the top by GeNorm and BestKeeper; however, it was ranked second by NormFinder. In the MeJA treatment subset, *SAM* was the most stable reference gene using NormFinder, but it was ranked second, third, or fourth in Delta Ct, Bestkeeper, and GeNorm, respectively. This apparent divergence is due to the nature of the different software programs and the different weights placed on different sources of variation [47]. GeNorm identified two reference genes with the highest degrees of similarity in terms of their expression profiles and the lowest intragroup variation [48,49]. Compared to the GeNorm algorithm, NormFinder software (https://www.moma.dk/normfinder-software/, accessed on 27 August 2022) is less robust with small sample sizes [50]. For BestKeeper, the standard deviation (SD) and coefficient of variation (CV) of the Ct values were calculated to determine the expression stability levels of the candidate reference genes. The RefFinder website is a popular tool for reference gene verification because it is free and allows a quick analysis to be performed; the website uses the three algorithms for reference gene validation starting from a single input of the Ct values only [51] and then calculates the mean weight of each gene by every algorithm [52]. A lower geomean rank implies that the reference gene is more stable. *SAM* was identified as the most stable reference gene for *U. rhynchophylla* in our experiments. Combining the analyses of several algorithms, an overall greater stability was obtained with *SAM* than with any other reference gene under all treatments. In addition, the stability levels of *PP2A* and *cdc73* were found to be good and only slightly lower than that of *SAM* in some algorithms, so they may be used as additional reference genes for qRT-PCR under similar experimental conditions in *U. rhynchophylla*. Specifically, we found that *TUA* and *EF-1β* were relatively unstable in terms of their expression levels in our experiments.

*U. Rhynchophylla* produces a large array of TIAs, yet they are generally produced at low production rates [3]. Many studies have shown that TIAs are defense molecules that are produced in response to biotic and abiotic stresses [53]. The biosynthesis pathway of TIAs in *U. Rhynchophylla* is a multistep, branched pathway [31]. Several genes encode the ~20 enzymes involved in the biosynthesis pathway of TIAs (Figure 1). Plant metabolic pathways are relatively complex, are usually multistep reactions, and each step may involve multiple enzymes, so modifying a single enzyme gene may sometimes make it difficult to achieve ideal results. Transcription factors, through the coordination of multiple genes, have the potential to significantly improve the expression of key synthetic enzyme genes and the production of pharmacological components and have become a research hotspot. Recently, some studies have reported that WRKY can simultaneously induce the coexpression of one or more genes or even the whole biosynthetic pathway [54,55,56]. Specific TFs can coordinate the transcription of multiple biosynthesis pathway genes, making them effective for use in metabolic engineering [22]. Some studies have shown that the W-box (TTGACC/T) is a cognate binding site for WRKY TFs. In this study, we found that the expression pattern of *TDC* is similar to that observed for *WRKY1* under exposure to ETH treatments. This could imply that *WRKY1* is possibly involved in its regulation. Analysis of the promoter sequence of *TDC* using PlantCARE (http://sphinx.rug.ac.be:8080/PlantCARE/, accessed on 11 September 2022) software to predict the promoter elements showed that *TDC* has the W-boxes necessary for WRKY TF binding. However, the *G8H* promoter was also found to possess a W-box, and its expression pattern is significantly different from that of *WRKY1*, suggesting that other WRKY TFs could regulate its expression.

## 4. Methods and Materials

### 4.1. Plant Material and Stress Treatment

Tips of *U. rhynchophylla* with a similar stem size (3 cm) were cut from plants growing in the medicinal plant garden scientific research base, Nanning City, Guangxi Province, China (108.19° E; 22.49° N). We obtained permission to collect *U. rhynchophylla*. The materials were preserved in the germplasm resource nursery of the Academy of Horticulture, Hunan Agricultural University and were identified by Prof. Shugen Wei. For abiotic and hormone treatments, tissue culture seedlings with nearly 15 leaves were grown for 60 days in tissue culture vessels. Tissue culture seedlings were placed in a chamber with a mean temperature of 25.0 ± 1.0 °C, a relative humidity of 60 ± 10%, and a day/night cycle of 12/12 h. *U. rhynchophylla* seedlings were subjected to three experimental treatments following previously reported methods [57,58]. A total of 12 equally growing plants were divided into four groups, including the following experimental groups: low temperature, ethylene (ETH), and methyl jasmonate (MeJA)-treated leaves, and a control group (distilled-water-treated leaves). For the hormone treatments, the plants were subjected to 100 μM MeJA or 100 μM ETH for 0, 1, 2, 4, 8, and 24 h. MeJA and ETH were customized and purchased from Colaber (Beijing, China). For cold stress, tissue culture seedlings were placed in a chamber at 4 °C for 0, 4, 8, 12, 24, and 48 h. The control group received no treatment but was watered with distilled water over time. All groups involved three biological replicates. The clipped leaves were washed with sterilized water, blotted dry, snap frozen in liquid nitrogen, and stored in a −80 °C freezer.

### 4.2. Total RNA Extraction and cDNA Synthesis

Samples consisting of 50–100 mg of tissue in liquid nitrogen were fully milled to a fine powder, and total RNA was extracted using the *SteadyPure* Plant RNA Extraction Kit (Accurate Biology, Changsha, China). The purity and concentration of the RNA were measured using a Micro Drop (BIO-DL, Shanghai, China), and the RNA integrity was checked on 1% agarose gels. The total RNA (500 ng) was used for reverse transcription with an *Evo M-MLV* RT Mix Kit with gDNA Clean (Accurate Biology, Changsha, China) to remove genomic DNA contamination for qPCR in a 20 μL volume according to the manufacturer’s instructions.

### 4.3. Selection of Candidate Genes, Primer Design, and RT-qPCR Conditions

Ten candidate reference genes were selected from the full-length genome database obtained using SMRT (Single-Molecule Real-Time) technology on the Oxford Nanopore PromethION (Unpublished, Benagen, Wuhan, China). Specifically, the data used for analysis were based on a “Genome + Full-length transcriptome + Metabolomics” coexpression analysis, in which upstream pathway genes in the alkaloid biosynthesis pathway of *U. rhynchophylla* were screened. Specific primer pairs were designed using Beacon Designer 7 software and restricted to primer sequences of 18–25 bp, amplicon lengths of 80–150 bp, melting temperatures (Tm) of 55–60 °C, and GC contents of 45–55% (Appendix A). The specificity of all primers was measured by dissociation curves. The cDNA template stock solution of the samples under different treatments was mixed in equal amounts and diluted five times, setting five concentration gradients, which were 5^0^, 5^−1^, 5^−2^, 5^−3^, and 5^−4^ times the template stock solution. The Ct values of the reference genes and target genes involved in the TIA pathway were obtained by qRT-PCR, and standard curves were generated to calculate the slope (k), correlation coefficient (R^2^), and amplification efficiency of the reference genes using the formula E = (10^−1/k−1^) × 100%. The reaction was carried out in 96-well plates with a Thermal Cycler Dice^TM^ Real Time System III (Takara, Kusatsu-shi, Japan). The 20 µL reaction system included the following components: 2 × 10 μL SYBR Green Pro Taq HS premix (Vazyme, Nanjing, China), 2 μL cDNA (100 ng/μL), 0.4 μL each of a forward primer and reverse primer (10 μM), and 7.2 μL RNase free water. The amplification reaction procedure was as follows: 95 °C, 30 s; 95 °C, 5 s; and 60 °C, 30 s for 40 cycles. Each RT-qPCR analysis was performed with three technical replicates and three biological replicates.

### 4.4. Data Analysis of the Reference Gene Expression Stability

GraphPad Prism 9 software was used to draw box plots of the distribution of the cycle threshold (Ct) of the ten candidate reference genes in all samples. Three statistical programs, GeNorm, NormFinder, and BestKeeper, and one algorithm, Delta Ct, were used to evaluate the stability levels of the candidate reference genes under all experimental treatments. Before using GeNorm and Normfinder software, the Ct values were converted to Q values [59] for subsequent analysis. The transformation formula used was Q = 2Ct^min^·− Ct^sample^ (Ct^sample^ is the Ct value of the candidate reference gene in different experimental groups, and Ct^min^ is the minimum Ct value of the candidate reference gene in all experimental groups). Bestkeeper and Delta Ct could be used to perform a computational analysis directly from the Ct values of the candidate genes. The difference is that it determines the stability of the reference genes based on the coefficient of variation (CV) and the standard deviation (SD) of the Ct values. Lower SD and CV values represent higher gene stability levels. Based on the results of the above software program, RefFinder (http://blooge.cn/RefFinder/, accessed on 23 September 2022) was used to perform a comprehensive stability ranking analysis. The expression levels of the *WRKY1* transcription factors and key enzyme genes in the upstream pathways of TIA synthesis (indole pathway and iridoid pathway) were analyzed under three stress treatments in *U. rhynchophylla* based on the selected reference gene, and the change pattern for the relative expression of each key enzyme gene under different treatments was calculated using the 2^−△△Ct^ method [60]. The significance difference analysis was calculated using Graphpad Prism 9.0 (two-way ANOVA of data). And, an analysis of the promoter sequences of the target genes using the PlantCARE (http://sphinx.rug.ac.be:8080/PlantCARE/, accessed on 11 September 2022) software was conducted to predict the promoter elements.

## 5. Conclusions

This research selected and evaluated the stability of ten candidate reference genes for RT-qPCR normalization under three experimental treatments in *U. Rhynchophylla* using tissue-culture-derived seedling leaves. Based on our analysis, *SAM* and *cdc 73* are the reference genes for normalizing expression levels in *U. Rhynchophylla* under our experimental conditions and are potentially optimal genes for normalization for other RT-qPCR experiments in *U. Rhynchophylla*. Using *SAM* and *cdc 73* as reference genes in the RT-qPCR analysis of the *WRKY1* and 14 genes associated with TIA biosynthesis showed that the expression pattern of *TDC* is similar to that of *WRKY1* under exposure to ETH treatment, implying that WRKY1 is involved in the regulation of *TDC* expression.

## Figures and Tables

**Figure 1 ijms-24-16330-f001:**
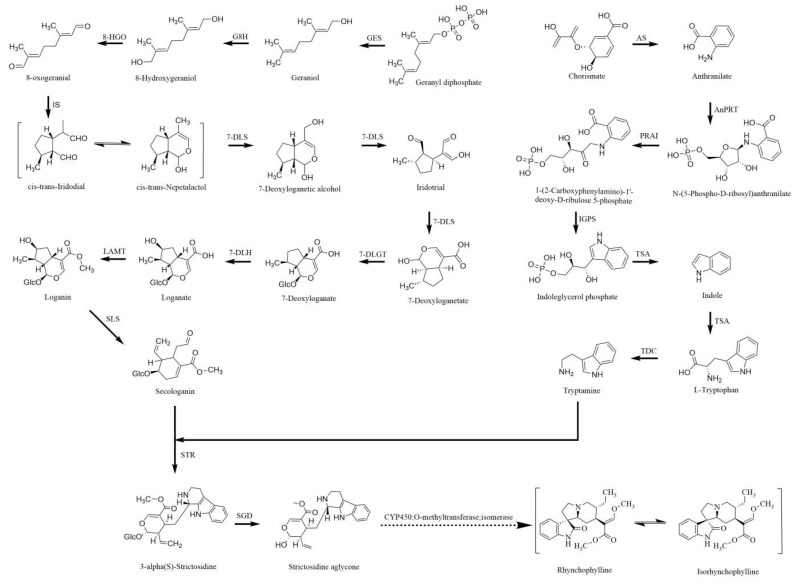
The biosynthetic pathway of rhynchophylline and isorhynchophylline in *U. rhynchophylla.* Solid arrows represent genes identified in the pathway and dashed arrows represent presumed genes.

**Figure 2 ijms-24-16330-f002:**
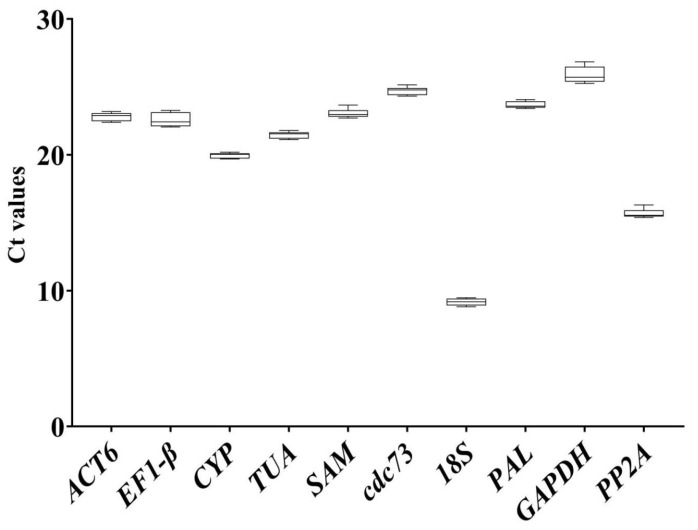
The Ct values of 10 candidate reference genes in all samples. The expression data are displayed as the Ct value of each reference gene in the sample of *U. rhynchophylla.* The boxplot indicates the 25th/75th percentiles and it contains the mean, maximum, and minimum values.

**Figure 3 ijms-24-16330-f003:**
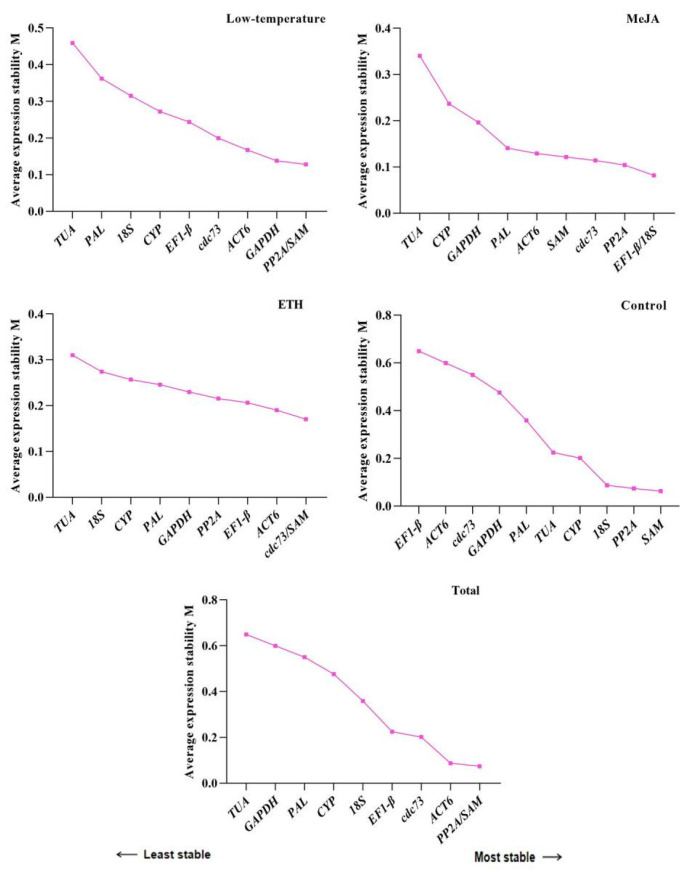
Average expression stability values (M) of the ten candidate reference genes using GeNorm. The expression stability was evaluated in samples from leaves of *U. rhynchophylla* subjected to low temperatures, MeJA (100 μmol), ETH (100 μmol), control (not treated), and total (all treated samples). The least stable genes are on the left with higher M values and the most stable genes are on the right with lower M values.

**Figure 4 ijms-24-16330-f004:**
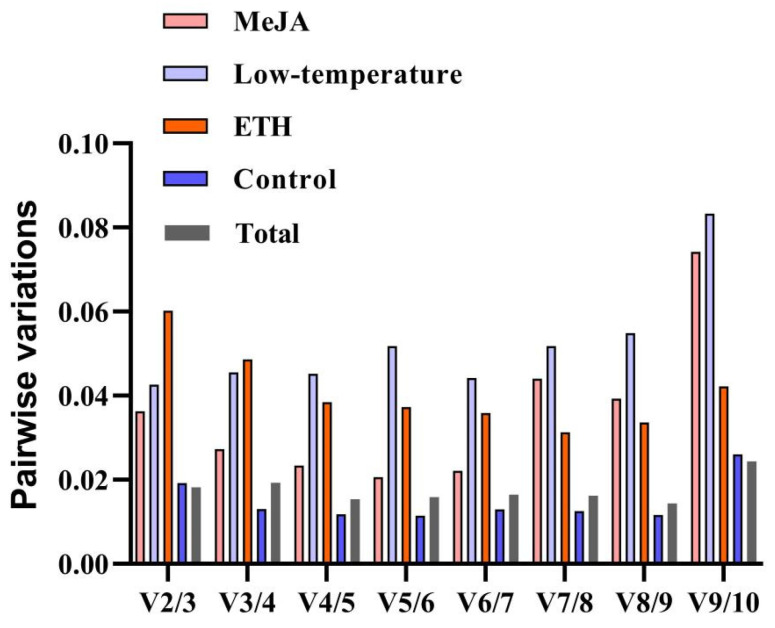
The pairwise variation (Vn/n + 1) scores of ten candidates measured by GeNorm. Different treatments are identified by different colors. The value used to determine the appropriate number of reference genes for RT-qPCR normalization is 0.15.

**Figure 5 ijms-24-16330-f005:**
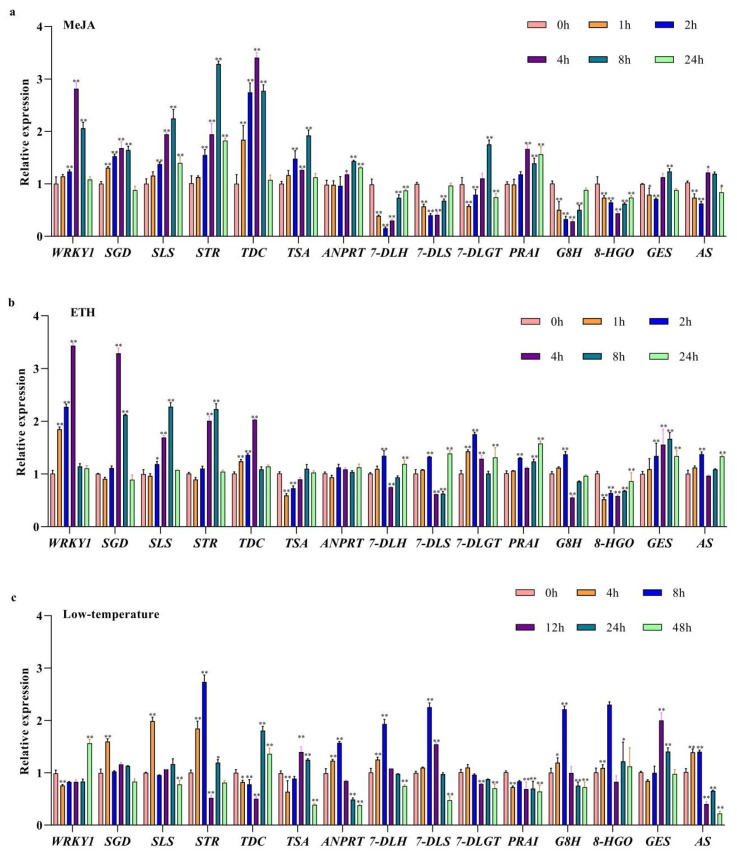
The relative expression of the *WRKY1* transcription factor and key enzyme genes in response to MeJA, ETH, and low temperatures. Different colors represent different times. (**a**) MeJA treatment; (**b**) ethylene treatment; (**c**) low-temperature treatment. The error bars represent the mean ± SD from three biological replicates, and asterisks indicate statistically significant differences compared with the controls (0 h).* *p* < 0.05; ** *p* < 0.01.

**Table 1 ijms-24-16330-t001:** Primer sequences, amplification efficiency, and correlation coefficient of the candidate reference genes of *U. rhynchophylla*.

Gene Abbreviation	Gene Name	Primer Sequence 5′-3′ (F/R)	Amplification Efficiency (%)	R^2^
*GAPDH*	Glyceraldehyde-3-phosphate dehydrogenase	TACCACCAACTGTCTTGCTCTT CGCCTCTCCAGTCCTTCATT	99.7	0.991
*CYP*	Cyclophilin	CGAGAAAGGCGTGGGAAAG TGAGACCCGTTGGTGTTGG	102.7	0.998
*PP2A*	Protein phosphatase 2A	CGCTGATGTTCTACGTCTACCTAAT AGTAGTAACGGTCCTCGGCTATA	99.8	0.996
*SAM*	S-adenosylmethionine decarboxylase	CACAATCTGGCATACGAAA AACTCACTTGGCTGGAAAC	102.8	0.998
*TUA*	α-Tubulin	TCCCTTCTTGAGCACACTGAT CCATCAAACCTCAAAGACGCA	101.7	0.995
*ACT6*	Actin 6	ACCGAGCGTGGTTATTCTT TTCCTGCTGCTTCCATTCC	99.3	0.995
*PAL*	Phenylalanine Ammonia lyase	ATCGCTGAATCCTCCAATA CCACCCTACTCCACAATACTT	102.8	0.993
*EF1-β*	Elongation factor 1 beta	AAGGCATCCACCAAGAAGA AAGGCAACAATGTCACAGC	105.0	0.996
*cdc73*	Cell division control protein 73	TGGTGGCTGTTTTCGTGTT TGATGCCGCTTATTCTTGC	100.2	0.995
*18S*	18S Ribosomal RNA	CTTCGGGATCGGAGTAATGA GCGGAGTCCTAGAAGCAACA	102.4	0.999

**Table 2 ijms-24-16330-t002:** Expression stability ranks of ten candidate housekeeping genes using NormFinder in *U. rhynchophylla*.

Rank	MeJA	ETH	Low Temperature	Control	Total
1	*SAM* 0.291	*SAM* 0.203	*PP2A* 0.104	*SAM* 0.145	*SAM* 0.507
2	*PP2A* 0.303	*ACT6* 0.282	*SAM* 0.260	*PP2A* 0.210	*cdc73* 0.605
3	*ACT6* 0.310	*cdc73* 0.427	*ACT6* 0.290	*CYP* 0.212	*CYP* 0.722
4	*cdc73* 0.334	*EF1-β* 0.443	*cdc73* 0.346	*18S* 0.237	*ACT6* 0.745
5	*18S* 0.434	*CYP* 0.547	*GAPDH* 0.541	*GAPDH* 0.253	*PAL* 0.813
6	*PAL* 0.513	*PP2A* 0.653	*18S* 0.542	*TUA* 0.273	*PP2A* 0.816
7	*EF1-β* 0.573	*GAPDH* 0.665	*CYP* 0.807	*PAL* 0.277	*EF1-β* 0.965
8	*CYP* 0.621	*PAL* 0.674	*PAL* 0.865	*cdc73* 0.333	*GAPDH* 0.965
9	*GAPDH* 0.753	*18S* 0.958	*EF1-β* 1.097	*ACT6* 0.341	*18S* 1.289
10	*TUA* 1.194	*TUA* 0.967	*TUA* 1.413	*EF1-β* 0.521	*TUA* 1.403

**Table 3 ijms-24-16330-t003:** Ranks of ten candidate reference genes in *U. rhynchophylla* (coefficient of variation (CV) ± standard deviation (SD)).

Rank	MeJA	ETH	Low Temperature	Control	Total
1	*ACT6* 1.32 ± 0.31	*SAM* 1.55 ± 0.35	*SAM* 1.95 ± 0.50	*SAM* 0.62 ± 0.14	*SAM* 1.98 ± 0.46
2	*cdc73* 1.40 ± 0.35	*ACT6* 2.01 ± 0.44	*PP2A* 2.06 ± 0.51	*PP2A* 0.87 ± 0.22	*cdc73* 2.17 ± 0.54
3	*SAM* 1.48 ± 0.33	*cdc73* 2.28 ± 0.57	*cdc73* 2.29 ± 0.53	*CYP* 0.88 ± 0.18	*ACT6* 3.03 ± 0.69
4	*PP2A* 1.55 ± 0.36	*PAL* 2.36 ± 0.55	*ACT6* 2.45 ± 0.55	*PAL* 0.94 ± 0.22	*PP2A* 3.38 ± 0.78
5	*EF1-β* 1.67 ± 0.25	*PP2A* 3.09 ± 0.51	*GAPDH* 2.66 ± 0.42	*TUA* 1.02 ± 0.22	*EF1-β* 3.40 ± 0.75
6	*PAL* 1.70 ± 0.38	*GAPDH* 3.16 ± 0.77	*18S* 2.95 ± 0.27	*cdc73* 1.04 ± 0.26	*CYP* 3.80 ± 0.75
7	*GAPDH* 2.67 ± 0.65	*CYP* 3.20 ± 0.62	*EF1-β* 2.97 ± 0.68	*GAPDH* 1.12 ± 0.17	*GAPDH* 3.82 ± 0.96
8	*CYP* 3.15 ± 0.63	*EF1-β* 3.25 ± 0.69	*PAL* 3.12 ± 0.75	*ACT6* 1.26 ± 0.29	*PAL* 4.26 ± 0.67
9	*18S* 3.27 ± 0.30	*TUA* 3.74 ± 0.76	*CYP* 3.66 ± 0.74	*EF1-β* 1.85 ± 0.42	*TUA* 5.36 ± 1.15
10	*TUA* 4.24 ± 0.94	*18S* 7.96 ± 0.87	*TUA* 5.23 ± 1.14	*18S* 2.02 ± 0.18	*18S* 9.17 ± 0.89

## Data Availability

The datasets generated and analyzed during the current study are available from the NCBI repository [*GES*:OP669346,*G8H*:OP669347,*8-HGO*:OP669348,*7-DLS*:OP669349,*7-DLGT*:OP669350,*7-DLH*:OP669351,*SLS*:OP669352,*SGD*:OP669353,*STR*:OP669354,*AS*:OP669355,*AnPRT*:OP669356,*PRAI*:OP669357,*TSA*:OP669358,*TDC*:OP669359,*WRKY1*:OP669360]. Appendix A, which supports the conclusions and description of the complete protocol, is included within the article. The gene sequence information mentioned in this study could be found from the literature based on the gene list in Appendix A.

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
