# Peer review of "Evaluation of Reference Genes for Normalizing RT-qPCR and Analysis of the Expression Patterns of WRKY1 Transcription Factor and Rhynchophylline Biosynthesis-Related Genes in Uncaria rhynchophylla"

_ijms, 2023, doi:10.3390/ijms242216330_

Round 1
Reviewer 1 Report
Comments and Suggestions for Authors
Dear authors, I appreciate the time taken to conduct this analysis. This is quite straight-forward work, however, to get published in a high-impact factor journal like IJMS, I recommend conducting a more extensive analysis. Please find below some of my major comments:
· The authors have mentioned that they have chosen the genes based on unpublished data. This is okay, but I request the authors to at least show something, based on which the reviewers can assess the basis of your selection. For e.g., Ubiquitin is one of the most common reference genes in plants, which I don’t find in the list.
· Also, I suggest screening more genes. At least up to 15 genes.
· The authors have used cDNA 2 μL (100 ng/μL), which corresponds to 200ng. Isn’t this too much, especially for HK genes, where it is expected to be expressed at even low temperatures?
Reviewer 2 Report
Comments and Suggestions for Authors
The work presented by Mu et al., entitled “Evaluation of Reference Genes for Normalizing RT‒qPCR and 2 Analysis of the Expression Patterns of WRKY1 Transcription 3 Factor and Rhynchophylline Biosynthesis-Related Genes in 4 Uncaria rhynchophylla” describes the gene expression analysis of housekeeping genes and certainly gives a new idea of selecting a reference gene in Uncaria rhynchophylla.
Authors are suggested to make following major corrections before it gets finally accepted-
1- Introduction part needs to be more emphasized on reference genes and expression analysis followed by Rhynchophylline biosynthesis pathway.
2- Discussion parts need to be more described with recent references.
Round 2
Reviewer 1 Report
Comments and Suggestions for Authors
The manuscript has been substantially improved. I am satisfied with the explanations provided by the authors. However, there are some places within the manuscript where the gene names are not italicized. Please check it throughout the manuscript. Overall, I recommend the manuscript for publication in the IJMS, after this minor correction. Cheers, and Good luck.!!!
Author Response
Please see the attachment! Thank you1

Reviewer 2 Report
Comments and Suggestions for Authors
Authors have incorporated the suggestions. This can be accepted now in this form.
Author Response
Thank you!